# Molecular Cloning and Identification of NADPH Cytochrome P450 Reductase from *Panax ginseng*

**DOI:** 10.3390/molecules26216654

**Published:** 2021-11-03

**Authors:** Xian Zou, Yue Zhang, Xu Zeng, Tuo Liu, Gui Li, Yuxin Dai, Yuanzhu Xie, Zhiyong Luo

**Affiliations:** Department of Biochemistry and Molecular Biology, School of Life Sciences, Central South University, Changsha 410008, China; 172511011@csu.edu.cn (X.Z.); zhang1045242781@126.com (Y.Z.); xuzeng_katharine@163.com (X.Z.); lt1994@csu.edu.cn (T.L.); ligui20061029@126.com (G.L.); yuxin_dai@csu.edu.cn (Y.D.); 172511004@csu.edu.cn (Y.X.)

**Keywords:** NADPH-cytochrome P450 reductase, ginsenoside, methyl jasmonate, prokaryotic expression, PgCPR1 and PgCPR2

## Abstract

Ginseng (*Panax ginseng* C.A. Mey.) is a precious Chinese traditional medicine, for which ginsenosides are the most important medicinal ingredients. Cytochrome P450 enzymes (CYP450) and their primary redox molecular companion NADPH cytochrome P450 reductase (CPR) play a key role in ginsenoside biosynthesis pathway. However, systematic studies of CPR genes in ginseng have not been reported. Numerous studies on ginsenoside synthesis biology still use *Arabidopsis* CPR (AtCPR1) as a reductase. In this study, we isolated two CPR genes (*PgCPR1*, *PgCPR2*) from ginseng adventitious roots. Phylogenetic tree analysis showed that both PgCPR1 and PgCPR2 are grouped in classⅡ of dicotyledonous CPR. Enzyme experiments showed that recombinant proteins PgCPR1, PgCPR2 and AtCPR1 can reduce cytochrome c and ferricyanide with NADPH as the electron donor, and PgCPR1 had the highest enzymatic activities. Quantitative real-time PCR analysis showed that PgCPR1 and PgCPR2 transcripts were detected in all examined tissues of *Panax ginseng* and both showed higher expression in stem and main root. Expression levels of the PgCPR1 and PgCPR2s were both induced after a methyl jasmonate (MeJA) treatment and its pattern matched with ginsenoside accumulation. The present investigation suggested PgCPR1 and PgCPR2 are associated with the biosynthesis of ginsenoside. This report will assist in future CPR family studies and ultimately improving ginsenoside production through transgenic engineering and synthetic biology.

## 1. Introduction

*Panax ginseng* C. A. Meyer is a traditional Chinese herbal medicine [1,2]. Ginsenoside is the most important medicinal active component in ginseng [3]. More and more research has found that ginsenoside can reduce oxidative stress response, protect the heart [4], slow down depression [5], prevent and treat Alzheimer’s disease [6,7,8], inhibit the growth of tumor cells [9,10,11,12] and other pharmacological effects. However, ginseng, a crop with a long growth period and a continuous cropping obstacle are seriously limited by the growing environment in yield and quality. Meanwhile, the contents of ginsenosides with unique pharmacological activities are very few [13]. So, the output of ginsenoside is far from meeting the market demand. Therefore, many researchers attempt to use functional gene identification and synthetic biology to improve the yield of ginsenoside.

In recent years, research into the ginsenoside biosynthesis pathway and its key enzymes have made continuous progress. Its biosynthesis pathway can be generally divided into three stages: Step one is to synthesize isoprene pyrophosphate (IPP) through the MVA pathway (Mevalonate pathway) and the MEP pathway (Methylerythritol 4-phosphate pathway); step two is to synthesize 2,3-oxidosqualene, which is the precursor of triterpenoid saponin; step three is the post-modification of compounds involved in the cyclization, hydroxylation, and glycosidation of 2,3-oxidosqualene [14,15,16,17] (Appendix A).

Many enzymes, such as SQE, DDS and CAS, involved in biosynthesis of ginsenosides are shown to be CYP450, which determine the diversity of ginsenoside monomers [18,19,20]. NADPH-cytochrome P450 reductases (CPR) is the redox molecular companion of the cytochrome P450 enzyme system, belonging to the flavin protein family. It is responsible for transferring the electrons in reductive NADPH via flavin mononucleotide (FMN) and flavin adenine dinucleotide (FAD) cofactors to the ferrous erythrocyte group of cytochrome P450 enzyme, providing electrons for a series of oxidation reactions involving cytochrome P450 enzyme [21]. Plant cytochrome P450 reductase-cytochrome P450 enzyme electron transfer system plays an important role in plant growth, development, biosynthesis of natural products and resistance to external stimuli [22].

In the plant genome, CYP450 is a superfamily of heme mercaptan protein, but CPRs are only encoded by a small number of genes. The amino acid sequences of CPR in different higher plants are highly homologous, so CPRs from different plants can be used as a molecular chaperone of P450 in yeast or other plants. However, different CPRs have different enzyme activities for the same substrate or P450 monooxygenase [23,24]. Sangkyu et al. showed that OsCPR2 showed the highest catalytic activity among the three rice CPRs [25]. In recent years, there are more and more CPRs of non-model plants. Two kinds of CPR were isolated and examined in cotton. The results showed that mechanical injury and inducer could induce the expression of GhCPR2, not GhCPR1 [26]. The sequence and function of ginseng cytochrome P450 have been analyzed one after another, but up to now, there has been no systematic study on ginseng CPR. *Arabidopsis* CPR (AtCPR1) is still used as a reductase in the study of CYP450 function and ginsenoside synthesis biology.

In this study, two CPR gene sequences PgCPR1(GenBank: MT160178) and PgCPR2(GenBank: MT160179) were selected from the MeJA-induced ginseng root transcription database and bioinformatics analysis was performed. The catalytic activities of PgCPR1, PgCPR2 and AtCPR1 were compared through an enzyme activity test. The expression patterns of PgCPR1 and PgCPR2 under different MeJA treatments were detected by qPCR. The systematic analysis of ginseng CPR is worthwhile in order to deepen our understanding of the CPR family and provide a reference for the study of ginseng secondary metabolites, and a more suitable CPR gene for the study of ginsenoside biosynthesis. This is of great significance for the construction of a suitable microbial expression system and the large-scale industrial production of ginsenosides.

## 2. Results

### 2.1. Molecular Cloning and Bioinformatics Analysis of PgCPR1 and PgCPR2 in Panax ginseng

In order to study the biosynthesis of ginsenosides, we performed transcriptome sequencing on *Panax ginseng* hairy roots and obtained the transcriptome information of ginseng. Two candidate CPR genes were screened from the transcriptome database and were named *PgCPR1* and *PgCPR2*. According to the information of the gene sequence, we designed specific primers to clone *PgCPRs* (Appendix A). The relative molecular weight of PgCPR1 is 78.56 kDa, and that of PgCPR2 is 75.47 kDa. Blast results show that PgCPR1 and PgCPR2 are 99% similar to ginseng CPR1 (AHA50098.1) and ginseng CPR2 (AIC73829.1), which have been uploaded into the NCBI database. The NCBI conserved domains finder results show that both CPRs contain all functional domains that bind to CYP450 and interact with coenzyme NADPH, FMN, and FAD (Appendix A). TMHMM prediction results show that both PgCPR1 and PgCPR2 contain transmembrane anchored domains (Appendix A).

Comparing the amino acid sequence of ginseng PgCPRs with the known CPRs amino acid sequence of model plants, PgCPRs have the same functional characteristics as CPRs in other plants (Figure 1). According to their N-terminal sequence, plant CPRs are classified into class I and class II [27]. Class I includes members of *Arabidopsis* AtCPR1 and *Andrographis* APCPR1, and Class II includes AtCPR2, APCPR2 and APCPR3. Here, multiple alignments show that, in the red box, class I is under a longer sequence, while class II has a shorter sequence (Figure 1). MEGA 7.0 program and the neighbor-joining method were used to analyze the phylogeny of the CPR amino acid sequence of ginseng and other plants. The results showed that PgCPR1 and PgCPR2 were both grouped in class II (Figure 2A). The CPR of class II can be induced by stress or induction and is more likely to participate in secondary metabolism. 

The three dimensional structure of the SWISS-MODEL showed (Figure 2B,C) that the similarity between PgCPR1 and ATR2 was 76.34%, and GMQE was 0.74 The similarity between PgCPR1 and ATR2 was 76.34%, and GMQE was 0.75 (Appendix A). Through bioinformatics analysis, it is found that the two CPRs of ginseng are very close to those of *Arabidopsis* CPR2. It is speculated that they may have the same redox function.

### 2.2. Heterologous Expression and Catalytic Activity Analysisof Recombinant PgCPRs and AtCPR1

The Coomassie blue staining results showed that the truncated PgCPR1, PgCPR2, and AtCPR1 were soluble after IPTG induction at low temperatures, and the purified CPR protein was obtained and used for subsequent enzyme activity experiments (Figure 3). Purified recombinant CPR protein was used for enzyme activity determination. The reaction of CPR protein was expected to be completed in a short time (3–5 min) [28,29]. In order to determine the demand of PgCPRs for electronic donors, NADH or NADPH dependent cytochrome c reduction activities of the recombinant proteins were determined respectively. For PgCPR1, PgCPR2 and AtCPR1, NADPH is effectively used as an electron donor to reduce cytochrome c. On the contrary, NADH cannot be recognized as an electron donor by them, which indicates that three CPRs especially use NADPH as an electron donor (Table 1). The kinetic properties of PgCPR1, PgCPR2 and AtCPR1 on cytochrome c, NADPH and ferricyanide were determined. Three CPRs activities of three different substrates all follow the typical Michaelis Menten curve (Figure 4). The catalytic efficiency of PgCPR1 to cytochrome c is almost the same as that of PgCPR2, which is higher than that of AtCPR1. For NADPH, relative catalytic activity (Kcat/Km value) of PgCPR1 is the highest, and that of AtCPR1 is the lowest. For K_3_Fe(CN)_6_, the relative catalytic activity of PgCPR1 is five times that of PgCPR2 and AtCPR1(Table 2). The purified PgCPR1 and PgCPR2 proteins have redox ability, which is higher than that of AtCPR1. In the future study of ginsenoside synthesis biology, using ginseng CPR instead of AtCPR1 may obtain a higher yield.

### 2.3. Expression Analysis of PgCPRs

Quantitative real-time PCR (qPCR) analysis was utilized to detect gene expression levels in different tissues. Transcripts of PgCPR1 and PgCPR2 exist in all examined tissues (Figure 5A), while PgCPR1 and PgCPR2 were preferentially expressed in the main root and stem. The expression of PgCPR2 was generally higher than that of PgCPR1 in all tissues.

As a plant hormone, MeJA [30] can induce ginsenoside synthesis [31]. The content of PgCPR 1 and PgCPR 2 in the callus was detected by qPCR after being induced in a concentration-dependent manner for 24 h. The results showed that the expression of PgCPR1 and PgCPR2 increased significantly with the increase of MeJA concentration and reached the maximum value in 100 μM MeJA treatments, and then decreased significantly in the 150 μM MeJA treatment. (Figure 5B). Therefore, we added 100 μM MeJA to the ginseng callus medium to study the expression level of PgCPR1 and PgCPR2 at different time points (0,4,8,12,24,48). The qPCRanalysis showed that PgCPR1 and PgCPR2 were slightly expressed before treatment, but are markedly increased to the highest point in 24 or 12 h and then decrease. The expression of PgCPR2 was higher than that of PgCPR1 at different induction times (Figure 5C). The results showed that PgCPR1 and PgCPR2 were induced by MeJA. It is speculated that PgCPR1 and PgCPR2 might be involved in ginsenoside synthesis.

## 3. Discussion

The current research shows that the cytochrome P450 is the key rate-limiting enzyme of ginsenoside synthesis. Previous reports have shown that CPR is the sole electron donor of all microsomes, and it is also the rate-limiting factor of P450 monooxygenase reaction [32,33]. In this study, we isolated two *Panax ginseng* cytochrome P450 reductase genes, *PgCPR1* and *PgCPR2*, and identified their protein structure, catalytic activity in vitro for cytochrome c and K_3_Fe(CN)_6_, and gene expression level. 

In this study, the comparison of two PgCPRs by bioinformatics analysis revealed an 85% similarity in their amino acid sequence. CPR contains several functional domains that are relatively conservative, but N-terminal sequences vary greatly [34]. Using the CPR amino acid sequence of the truncated N-terminal sequence, we can construct phylogenetic trees that are similar to using the full-length sequence [27,35]. The truncation of the N-terminal transmembrane region can increase the solubility of CPR protein but will remain unchanged in its catalytic activity. We conjectured that the N-terminal transmembrane region might be responsible for P450s coupling rather than the reductase activity. Conserved domains in other regions were related to NADPH dependent cytochrome P450 reductase activity. 

Based on sequence comparison, PgCPR1 and PgCPR2 belong to class II. Ro et al. believe that CPR class I in general has a short N-terminal, but CPR class II invariably contains a longer N-terminal [27]. Nevertheless, PgCPR2 with N-terminal shorter than PgCPR1 is classified as class II. Amusingly, CPR2 in hybrid poplar, classified as class II, also has a short N-terminal [27]. *Andrographis paniculata* CPR3 has a very short N-terminal, which is also classified as class II [29]. In their research, Ro et al. think hybrid poplar CPR2 is an exception. According to our results, this appears to be no exception, although rarely seen. Cloning more CPR in other species may lead to more class II CPR with short N-terminal. So, we think that the length of N-terminal is an essential standard, but it is not the unique criterion divided into two classes. Additional sequence information is equally significant for the classification of the two classes.

MeJA is a plant-specific signal regulator, which has been proven to be a valid activator for the biosynthesis of secondary metabolites of several medicinal plants [36]. In ginseng, the up-regulation of critical enzyme genes, for instance, PgCYP450 after MeJA induction led to the accumulation of ginsenoside [37,38,39]. Both CPRs from *Panax ginseng* were induced after MeJA induction, but PgCPR2 was significantly higher than PgCPR1. The role of MeJA and its correlation with ginsenoside production connote that PgCPRs may play an important role in ginsenoside synthesis. In future experiments, we plan to use transgenic technology to over express and silence PgCPR1 and PgCPR2 genes, which will further clarify their functions in ginsenoside synthesis. The identification of CPRs of *Panax ginseng* is helpful for illustrating the function of P450 monooxygenase and the biosynthesis pathway of ginsenoside.

The endogenous CPR of yeast is not compatible with plant P450s [40]. Therefore, the WAT11 strain containing AtCPR1 is usually used to validation the function of P450s in different plants [41,42]. The functional plant CPRs can provide electrons for many CYP450 of many different plants, but the matching of P450 with suitable CPRs can significantly improve the production of a secondary metabolite.While the PgCPR1, PgCPR2 and AtCPR2-1 were introduced into engineering yeast respectively, the PPD produced by PgCPR1 and PgCPR2 yeast is greater than twice that of AtCPR2-1 yeast [43]. When CPRs from various plants were paired with CYP450: Uni25647, the titers of 11-oxo-β-amyrin were increased by 1422 times, and the glycyrrhetinic acid and were increased by 945.5 times [44,45]. Although the amino acid sequence of CPRs in plants is highly homologous, their enzyme activities in the same substrate or CYP450 appear to be different. In the production of ginsenosides by yeast metabolic engineering, different CYP450 paired with an appropriate CPR can increase ginsenoside production significantly.

Acting in accordance with the above results, ginseng CPRs can be used in yeast engineering to produce secondary metabolites. Through the establishment of yeast cell factories, more optimization and feasible strategies are offered to improve the supply of ginsenoside. PgCPRs can be used as an ideal biological module in synthetic biology. It can be combined with the existing biological system to redesign or develop completely different ginsenoside production methods. In the future, we will combine PgCPR1 and PgCPR2 with PgDDS, PgCYPs and PgUGTs to determine the best match. This provides a basis for the construction and synthetic biology of ginsenoside yeast engineering bacteria.

## 4. Materials and Methods 

### 4.1. Plant Materials and Culture Conditions

The *Arabidopsis* cDNA used in this study was provided by Hunan University. The ginseng used was four-year-old ginseng from the Changbai Mountain Forest of Jilin Province. The callus and adventitious roots of ginseng were induced and subcultured in our laboratory, which laid a foundation for the subsequent experiments.

### 4.2. Total RNA Extraction and cDNA Synthesis

In this experiment, the total RNA of all plant tissues was extracted with the Tiangen RNA prep Pure Plant kit (Polysaccharides&Polyphenolics-rich) (Tiangen, Beijing, China). Then the concentration and purity of RNA were measured by a spectrophotometer (Thermo Fisher Scientific, Waltham, MA, USA). The first cDNA was synthesized with the HiScript^®^ II Reverse Transcriptase kit (Vazyme, Nanjing, China) and was stored at −20 °C for future use.

### 4.3. Isolation and Cloning of PgCPRs Coding Sequences

The primers used in this study are listed in the table (Appendix A). The specific primers were synthesized by the Invitrogen company. According to the PgCPRs gene sequence obtained from the transcriptome database of the laboratory, the target fragment was amplified by PrimeSTAR^®^ Max DNA Polymerase (Takara, Beijing, China). The PCR product was cloned into pCE2 TA/Blunt-Zero Vector by 5min^TM^ TA/Blunt-Zero Cloning Kit (Vazyme, Nanjing, China) and transformed into *E. coli* DH5 α. The recombinant plasmid was isolated and sequenced by Tsingke Biotechnology.

### 4.4. Bioinformatic Analysis of PgCPR1 and PgCPR2

Use ORF finder (https://www.ncbi.nlm.nih.gov/orffinder/ (accessed on 7 October 2021)) to find the Open Reading Frame of PgCPR2. The NCBI conserved domains finder (https://www.ncbi.nlm.nih.gov/Structure/cdd/wrpsb.cgi (accessed on 7 October 2021)) was used to find the conserved domain of DNA. Calculate assumed molecular weight using ExPASytool (http://ca.expasy.org/tools (accessed on 7 October 2021)). The protein transmembrane helices were predicted by TMHMM Server v. 2.0 (http://www.cbs.dtu.dk/services/TMHMM/ (accessed on 7 October 2021)). Amino acid residues of CPRs were compared with the ClustalW program. Then, DNAMAN software was employed for multiple sequence alignment analysis. Phylogenetic trees are connected by the neighbor-joining method using MEGA 7.0 software. Bootstrap analysis of 1000 replicates was also performed to obtain the level of confidence of each branch. Based on the *Arabidopsis* CPR2 (ATR2) protein as a three-dimensional homologous model, the third-order structure of ginseng CPR protein was predicted by using the SWISS-MODEL online tool (https://swissmodel.expasy.org/ (accessed on 7 October 2021)). 

### 4.5. Prokaryotic Expression and Protein Purification of Recombinant PgCPR Proteins 

All plant CPRs have nearly 60 amino acid transmembrane structures, which are located at the N-terminal. Considering the optimization of the solubility of prokaryotic expression protein, the coding sequence of PgCPRs without the N-terminal membrane-anchored domain was amplified. Using ClonExpress^®^ II one-step cloning Kit (Vazyme, Nanjing, China), the target sequence was inserted into the middle of the *Hind III* site and *EcoRI* site of expression vector pET28a, and the expression, purification and kinetic analysis of the binding protein were as described above [31]. After sequencing and validation, the recombinant vector was transformed into *E. coli BL21* (DE3) active cells (Qingke, Beijing, China). Positive colony in LB medium (containing 100 μg/mL Kana) at 37 °C and 200 rpm. When OD600 = 1.0, add IPTG to the final concentration of 1 mM and induce at 16 °C for 12 h. The cells were then collected and resuspended in a binding buffer (50mM NaH_2_PO_4_, 300 mM NaCl, 10mM imidazole, pH 8.0) and ultrasonically treated on ice for 5 s to 20 times. After centrifugation at 4 °C for 30 min at 12,000 rpm (13,201× *g*), the Ni-NTA resins were added to the supernatant for affinity purification. The protein samples were separated by SDS-PAGE and stained by Coomassie brilliant blue R250. The concentration of recombinant protein was determined by Pierce™ BCA Protein Assay Kit (Bio-Rad, Hercules, CA, USA).

### 4.6. Enzyme Activity Assays In Vitro

The enzyme activities of AtCPR1 and PgCPRs in vitro were determined. The experimental methods refer to the reported articles [27]. All incubation and reaction were carried out at 25 °C in 50 mmol/L Tris HCl buffers (pH7.4, containing 0.1 mmol/L EDTA). Cytochrome reduction was monitored by an increase in absorbance at 550 nm. The molar absorption coefficient of cytochrome c was 21 mM^−1^cm^−1^. The reduction of K_3_Fe(CN)_6_ was monitored at 424 nm (1.02 mM^−1^cm^−1^). To determine the kinetic parameters of cytochrome c, 100 μmol/L, NADPH was added to the reaction mixture containing different concentrations of cytochrome c (0–800 μmol/L). The kinetic parameters of NADPH were determined with cytochrome c solution of 100 μmol/L, the concentration range of NADPH was 0–400 μmol/L. The kinetic parameters of K_3_Fe(CN)_6_ were determined and 100 μmol/L NADPH was added to the reaction mixture containing different concentrations of K_3_Fe(CN)_6_ (0–660 μmol/L). The absorption changes with time were recorded on the epoch spectrophotometer (BioTek, Winooski, VT, USA). GraphPad Prism 5 software was used to carry out nonlinear regression analysis and calculation. At least three independent copies of each sample were tested.

### 4.7. Quantitative Real-Time PCR

In this paper, the main root, lateral root, stem, leaf, and fruit of *Panax ginseng* were collected, and the callus of *Panax ginseng* was induced by MeJA. Expression patterns of PgCPRs were detected by using the cDNA of different tissues or processing stages as the template of qPCR amplification. The real-time PCR was carried out on a Mastercycler ep realplex Real-time fluorescence quantitative PCR instrument (Eppendorf, Germany) by using a Bioteke miRNA qPCR detection kit (SYBR Green) miRNA (Bioteke, Beijing, China). According to the requirements of the real-time PCR system, the primers of the qPCR specific reference gene, PgCPR1 and PgCPR2 are listed in Appendix A. The calibration curve was drawn for each primer pair and was quantified by GraphPad prism 6 analysis software. Each sample was tested for at least three independent biological repeats.

## Figures and Tables

**Figure 1 molecules-26-06654-f001:**
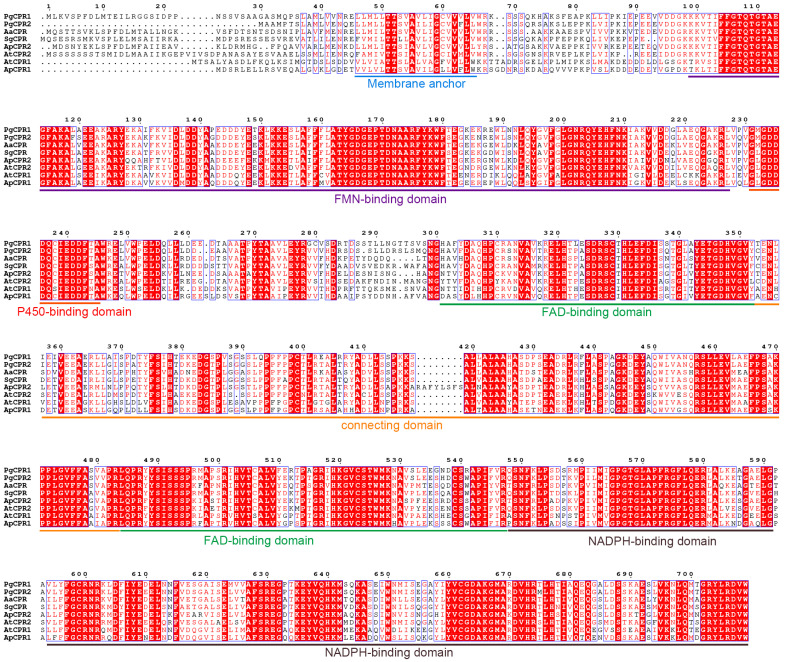
The amino acid sequence of PgCPRs was compared with that of other CPR homologs. The CPR for sequence alignment is from *Panax ginseng* (PgCPR1: MT160178, PgCPR2: MT160179), *Artemisia annua* (AaCPR: ABM88789), *Siraitia grosvenorii* (SgCPR: AYE89265), *Andrographis paniculate* (ApCPR1: AQT38168, ApCPR2: AQT38169) and *Arabidopsis thaliana* (AtCPR1: NP194183, AtCPR2: NP194750) by the program Clustal W. The conservative domains are underlined as follows: Membrane anchor (blue), FMN-binding domain (purple), P450-binding domain (red), connecting domain (aurantium), FAD-binding domain (green) and NADPH binding domain (brown).

**Figure 2 molecules-26-06654-f002:**
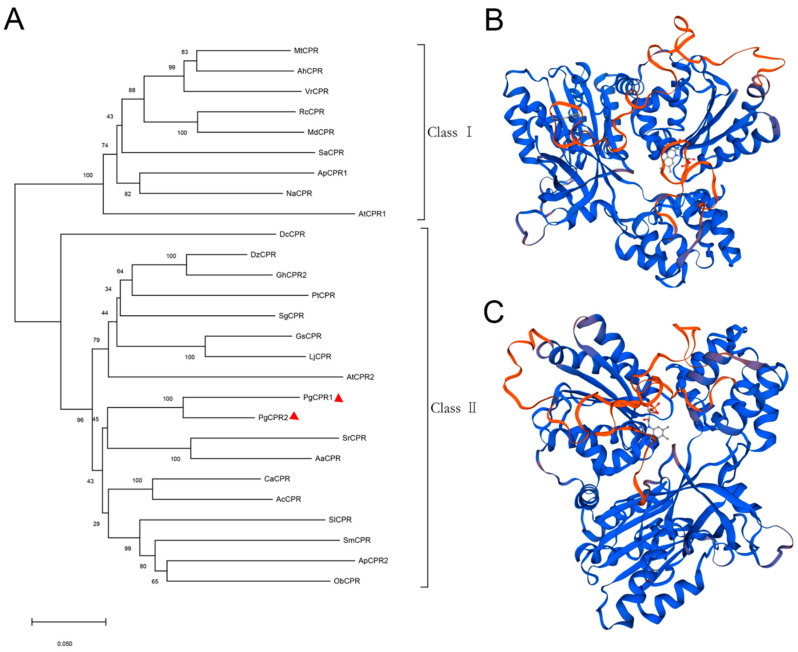
Phylogenetic analysis of PgCPRs. (**A**) The phylogenetic tree was constructed by the neighbor-joining method of MEGA 7.0, and 27 amino acid sequences of CPR protein were analyzed, which come from the following plant species: *Arachis hypogaea* (AhCPR: XP025668510), *Artemisia annua* (AaCPR: ABM88789), *Actinidia Chinensis* (AcCPR: PSS36215), *Andrographis paniculate* (ApCPR1: AQT38168, ApCPR2: AQT38169), *Arabidopsis thaliana* (AtCPR1: NP194183, AtCPR2: NP194750), *Camptotheca acuminate* (CaCPR: AJW67229), *Dendrobium catenatum* (DcCPR: PKU78870), *Durio zibethinus* (DzCPR: XP022729926), *Gossypium hirsutum* (GhCPR2: NP001314398), *Glycine soja* (GsCPR: KHN17598), *Lotus japonicus* (LjCPR: BAG68945 ), *Malus domestica* (MdCPR: XP028963051), *Medicago truncatula* (MtCPR: RHN70186), *Nicotiana attenuate* (NaCPR: OIT38777), *Ocimum basilicum* (ObCPR: ANW46529), *Panax ginseng* (PgCPR1: MT160178, PgCPR2: MT160179), *Populus trichocarpa* (PtCPR: XP002324469), *Rosa chinensis* (RcCPR: XP024171228), *Santalum album* (SaCPR: ANQ46483), *Siraitia grosvenorii* (SgCPR: AYE89265), *Solanum Lycopersicum* (SlCPR: XP004242931), *Salvia miltiorrhiza* (SmCPR: CBX24555), *Stevia rebaudiana* (SrCPR: ABB88839), *Vigna radiata* (VrCPR: NP001304239). The scale represents the phylogenetic distance calculated from the number of differences. (**B**,**C**), The three-dimensional structure diagrams of the PgCPR1 and PgCPR2, constructed by Swiss-Model software.

**Figure 3 molecules-26-06654-f003:**
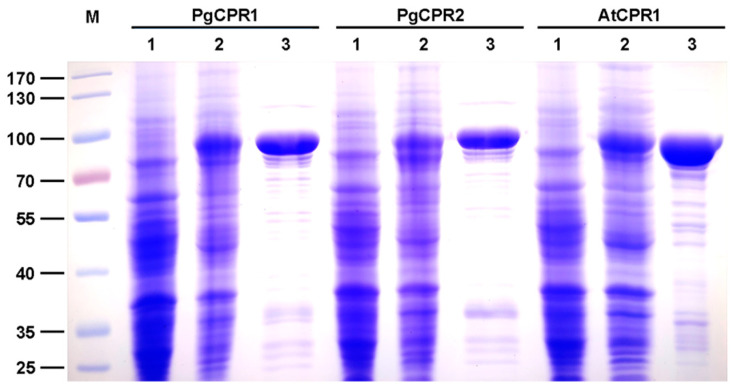
Expression and purification of recombinant PgCPRs and AtCPR1. The SDS-PAGE analysis of PgCPRs and AtCPR1 stained with Coomassie blue R-250. Lane M: protein marker; Lane 1: The crude enzyme in BL21 without IPTG. Lane 2: IPTG induced crude enzyme; Lane 3: The purified protein with his-tag.

**Figure 4 molecules-26-06654-f004:**
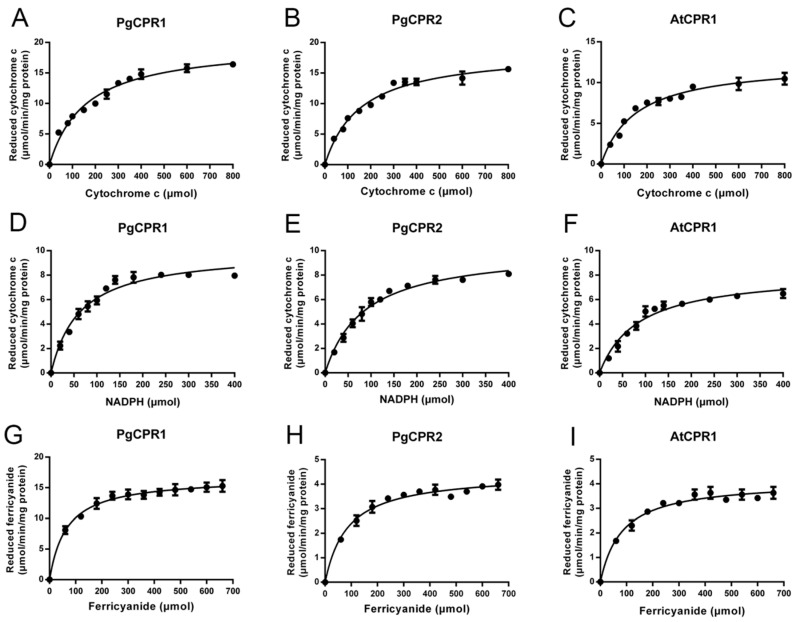
Kinetic parameters of the recombinant protein. (**A**–**C**), the kinetic parameters of recombinant protein to cytochrome c were measured by 550 nm absorbance increase method; (**D**–**F**), the kinetic parameters of NADPH were measured by cytochrome reduction method; (**G**–**I**) the kinetic parameters of K_3_Fe(CN)_6_ were studied by 424 nm absorbance change method.

**Figure 5 molecules-26-06654-f005:**
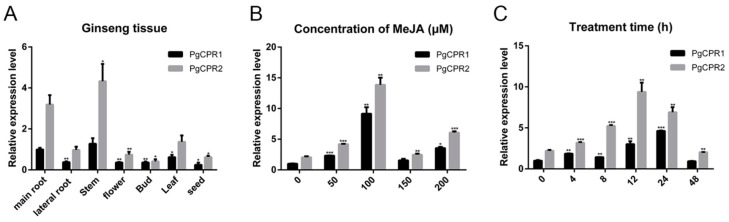
Expression analysis of PgCPRs. (**A**) The relative expression level of PgCPRs in different tissues of ginseng was calculated by using the expression level of PgCPR1 at the main root as standard 1. (**B**) The expression level of PgCPR1 at 0 μM MeJA was used as standard 1 to calculate the relative expression level of PgCPRs in different concentrations of MeJA at 24 h. (**C**) The expression level of PgCPRs at different time points was induced by 100 μM MeJA. The expression level of PgCPR1 at 0 h was used as the standard. All the above expression levels were standardized according to the expression level of β-actin. The standard deviation of three independent repeated tests is expressed by error bars. The asterisk indicates that there is significant difference in *t*-test, * *p* < 0.05, ** *p* < 0.01, *** *p* < 0.001.

**Table 1 molecules-26-06654-t001:** Specific activities of recombinant CPRs in reducing cytochrome c (100 μmol/L) with 100 μmol/L of NADPH or NADH. All data represent means ± standard deviations (SD). ND: Not detectable.

	Specific Activity (μmol/min/mg Protein)	
	NADPH	NADH
PgCPR1	7.23 ± 0.45	ND
PgCPR2	6.53 ± 0.37	ND
ATCPR1	5.56 ± 0.26	ND

**Table 2 molecules-26-06654-t002:** Kinetic analysis of recombinant CPRs. All data represent the mean ± SDS calculated by nonlinear regression analysis using GraphPad Prism 7.0 software.

		Vma (μmol/min/mg)	Km (μmol/L)	Kcat (min^−^^1^)	Kcat/Km
**Cytochrome c**	PgCPR1	19.85 ± 0.5344	159.4 ± 12.1	397 ± 10.688	2.49059
	PgCPR2	18.72 ± 0.4907	157.3 ± 11.69	374.4 ± 9.814	2.380165
	ATCPR1	12.49 ± 0.3651	151.9 ± 12.78	249.8 ± 7.302	1.644503
**NADPH**	PgCPR1	9.962 ± 0.2905	62.9 ± 5.807	199.24 ± 5.81	3.167568
	PgCPR2	10.14 ± 0.2611	85.27 ± 6.07	202.8 ± 5.222	2.378328
	ATCPR1	8.235 ± 0.2998	84.61 ± 8.548	164.7 ± 5.996	1.946578
**K3Fe(CN)6**	PgCPR1	16.67 ± 0.2791	64.91 ± 5.639	333.4 ± 5.582	5.136343
	PgCPR2	4.441 ± 0.08907	85.3 ± 7.601	88.82 ± 1.7814	1.041266
	ATCPR1	4.119 ± 0.093	82.73 ± 8.442	82.38 ± 1.86	0.995769

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
