# Peer review of "Molecular Cloning and Identification of NADPH Cytochrome P450 Reductase from Panax ginseng"

_molecules, 2021, doi:10.3390/molecules26216654_

Round 1
Reviewer 1 Report
Xian Zou and co-workers have carried out interesting and important research in the field of synthetic biology. Access to a protein from the cytochrome family, which is expressed by Panax Ginseng, has been opened.The obtained results are important for the study of electron transfer systems in plants. This article may be accepted for publication and will be greeted with interest by the readers.
Author Response
Dear Editors and Reviewers:
Thank you for your letter and for the reviewers’ comments concerning our
manuscript entitled “Molecular cloning and identification of NADPH cytochrome P450 reductase from Panax ginseng” (ID: molecules-1383173). Those comments are all valuable and very helpful for revising and improving our paper, as well as the important guiding significance to our researches.
Reviewer 2 Report
Review of a manuscript “Molecular Cloning and Identification of NADPH Cytochrome P450 Reductase from Panax Ginseng” by Xian Zou and coauthors submitted to “Molecules”
Panax ginseng is a traditional Chinese herbal medicine used to enhance physical performance and to increase resistance to stress and aging. It possesses multiple beneficial effects, including cardioprotection, reduction of depression, improvements in Alzheimer’s disease’s patients, slowing down the cancer cells growth, etc. An important active component in ginseng is Ginsenoside. The authors investigated the pathway of ginsenoside biosynthesis and established an important role of PgCPR1 and PgCPR2 in the biosynthesis of ginsenoside. This is an important study of the ginseng biochemistry, the results of which will be important for the readers of “Molecules”. The strong point of the manuscript is that the authors received new results indicating that PgCPR1 and PgCPR2 are associated with the biosynthesis of ginsenoside. The weak point is that some of the sentences in the manuscript are written in a bad style, difficult to understand which need to be rewritten in a more clear way.
The following corrections should be made:
Abstract:
Lines 10-11: “Panax ginseng is a precious Chinese traditional medicine, its most important medicinal
ingredient is ginsenoside, cytochrome P450 enzymes (CYP450) play a key role in ginsenoside biosynthesis pathway.” The meaning of this sentence is unclear, it should be corrected to become easily understandable.
Introduction:
Lines 34-35: “However, ginseng is a long growth period and continuous cropping obstacles..”
The sentence is clumsy and should be rewritten in an easy to understand way”.
Line 47-48:” Among them, CYP450s are necessary for the ginsenoside synthesis pathway and participates in a variety of oxidative metabolic reactions which determine the diversity of ginsenoside monomers [18-20]”. Bad English, the sentence needs correction.
Line 59: “The amino acid sequence of CPR in plants has high homology”. It is unclear, homology to what?
Lines 58-73: The authors present here details which are excessive for the Introduction. This date would be more appropriate for a review article. Should be truncated and presented in a more concise way.
Line 179: “As a plant hormone, MeJA can induce ginsenoside synthesis [32].”
This sentence should be written with an addition of a reference to MeJA as follows : ”As a plant hormone, MeJA [reference to: “Phytochemicals as regulators of genes involved in synucleinopathies. Biomolecules, 2021, 11 (5), 624] can induce ginsenoside synthesis [32].”
Legend to Figure 1. The authors should add to this figure legend a brief description of box1. Also P450-binding domain (orange) looks very similar to red color, so it should be marked by another color.
Discussion:
Line 203-204 “In this study, we isolated two Panax ginseng cytochrome P450 reductase genes, PgCPR1 and PgCPR2, and identified their gene structure, catalytic activity, and gene expression level.”. It sounds as P450 reductase genes possess catalytic activity. The authors should explain that they investigated catalytic activity of the proteins encoded by these genes.
Line 206. "In this study, the two PgCPRs have a high consistency (85%).” What the authors mean by consistency? Similarity, identity, homology? Also it is not a correct way to say that “In this study the two PgCPRs have a high consistency”. It should be rewritten in a way similar to that: “The comparison of two PgCPRs by … revealed the 85% similarity in their amino acid sequence”
Author Response
Dear Editors and Reviewers:
Thank you for your letter and for the reviewers’ comments concerning our manuscript entitled “Molecular cloning and identification of NADPH cytochrome P450 reductase from Panax ginseng” (ID: molecules-1383173). Those comments are all valuable and very helpful for revising and improving our paper, as well as the important guiding significance to our researches. We have studied comments carefully and have made correction which we hope meet with approval. Revised portion are marked in red in the paper.
Abstract:
Lines 10-11: “Panax ginseng is a precious Chinese traditional medicine, its most important medicinal ingredient is ginsenoside, cytochrome P450 enzymes (CYP450) play a key role in ginsenoside biosynthesis pathway.” were corrected as “Ginseng (Panax ginseng C.A. Mey.) is a precious Chinese traditional medicine, for which ginsenosides are the most important medicinal ingredients. Cytochrome P450 enzymes (CYP450) and their primary redox molecular companion NADPH cytochrome P450 reductase (CPR) play a key role in ginsenoside biosynthesis pathway.”
Introduction:
Lines 34-35: “However, ginseng is a long growth period and continuous cropping obstacles..” were corrected as “However, ginseng, a crop with a long growth period and continuous cropping obstacle are seriously limited by the growing environment in yield and quality.”
Line 47-48:” Among them, CYP450s are necessary for the ginsenoside synthesis pathway and participates in a variety of oxidative metabolic reactions which determine the diversity of ginsenoside monomers [18-20]”. were corrected as “Many enzymes such as SQE, DDS, CAS involved in biosynthesis of ginsenosides are shown to be CYP450, which determine the diversity of ginsenoside monomers[18-20].”
Line 59: “The amino acid sequence of CPR in plants has high homology. It can be used as a molecular chaperone of P450 from different plants.” were corrected as “The amino acid sequences of CPR in different higher plants are highly homologous, so CPRs from different plants can be used as a molecular chaperone of P450 in yeast or other plants.”
Lines 58-73: The authors present here details which are excessive for the Introduction. This date would be more appropriate for a review article. Should be truncated and presented in a more concise way.
Lines 58-73 were corrected as “However, different CPRs have different enzyme activities for the same substrate or P450 monooxygenase [23-24]. Sangkyu et al. showed that OsCPR2 showed the highest catalytic activity among the three rice CPRs [25]. In recent years, there are more and more CPRs of non-model plants. Two kinds of CPR were isolated and examined in cotton. The results showed that mechanical injury and inducer could induce the expression of GhCPR2, not GhCPR1 [26]. The sequence and function of ginseng cytochrome P450 have been analyzed one after another, but up to now, there is no systematic study on ginseng CPR. Arabidopsis CPR (AtCPR1) is still used as a reductase in the study of CYP450 function and ginsenoside synthesis biology.”
Line 179: “As a plant hormone, MeJA can induce ginsenoside synthesis [32].”
This sentence should be written with an addition of a reference to MeJA as follows : ”As a plant hormone, MeJA [reference to: “Phytochemicals as regulators of genes involved in synucleinopathies. Biomolecules, 2021, 11 (5), 624] can induce ginsenoside synthesis [32].”
Line 179 were corrected as “As a plant hormone, MeJA [30] can induce ginsenoside synthesis [31].”.
30 Phytochemicals as regulators of genes involved in synucleinopathies. Biomolecules, 2021, 11 (5), 624.
Legend to Figure 1. The authors should add to this figure legend a brief description of box1. Also P450-binding domain (orange) looks very similar to red color, so it should be marked by another color.
The boxⅠ is not a conserved domain, so I delete it. The P450-binding domain in Figure 1 is red color.
Discussion:
Line 203-204 “In this study, we isolated two Panax ginseng cytochrome P450 reductase genes, PgCPR1 and PgCPR2, and identified their gene structure, catalytic activity, and gene expression level.” were corrected as “In this study, we isolated two Panax ginseng cytochrome P450 reductase genes, PgCPR1 and PgCPR2, and identified their protein structure, catalytic activity in vitro for cytochrome c and K3Fe(CN)6, and gene expression level.”
Line 206. "In this study, the two PgCPRs have a high consistency (85%).” were corrected as “In this study, The comparison of two PgCPRs by bioinformatics analysis revealed the 85% similarity in their amino acid sequence.”